# Bread Wheat (*Triticum aestivum*) Responses to Arbuscular Mycorrhizae Inoculation under Drought Stress Conditions

**DOI:** 10.3390/plants10091756

**Published:** 2021-08-24

**Authors:** Neila Abdi, Angeline van Biljon, Chrisna Steyn, Maryke Tine Labuschagne

**Affiliations:** Department of Plant Sciences, University of the Free State, Bloemfontein 9300, South Africa; neilaabdi@gmail.com (N.A.); avbiljon@ufs.ac.za (A.v.B.); bothac@ufs.ac.za (C.S.)

**Keywords:** bread wheat, arbuscular mycorrhiza, drought stress, morphological traits, physiological traits, antioxidants

## Abstract

Abiotic constraints such as water deficit reduce cereal production. Plants have different strategies against these stresses to improve plant growth, physiological metabolism and crop production. For example, arbuscular mycorrhiza (AM)—bread wheat association has been shown to improve tolerance to drought stress conditions. The objective of this study was to determine the effect of AM inoculation on plant characteristics, lipid peroxidation, solute accumulation, water deficit saturation, photosynthetic activity, total phenol secretion and enzymatic activities including peroxidise (PO) and polyphenol oxidase (PPO) in two bread wheat cultivars (PAN3497 and SST806) under well-watered and drought-stressed conditions in plants grown under greenhouse conditions, to determine whether AM can enhance drought tolerance in wheat. AM inoculation improved morphological and physiological parameters in plants under stress. The leaf number increased by 35% and 5%, tiller number by 25% and 23%, chlorophyll content by 7% and 10%, accumulation of soluble sugars by 33% and 14%, electrolyte leakage by 26% and 32%, PPO by 44% and 47% and PO by 30% and 37% respectively, in PAN3497 and SST806, respectively. However, drought stress decreased proline content by 20% and 24%, oxidative damage to lipids measured as malondialdehyde by 34% and 60%, and total phenol content by 55% and 40% respectively, in AM treated plants of PAN3497 and SST806. PAN3497 was generally more drought-sensitive than SST806. This study showed that AM can contribute to protect plants against drought stress by alleviating water deficit induced oxidative stress.

## 1. Introduction

Drought stress is the single largest abiotic stress factor leading to reduced cereal (rice, wheat and maize) yields [1]. In 2013, approximately 65 million ha of wheat was affected by drought stress [2]. The impact of drought on wheat productivity is becoming increasingly important in the world, as wheat represents a large proportion of total consumed calories in most populated regions [3,4]. Yield potential is highly variable due to differences in the timing and intensity of stress in the growing cycle [5]. Grain yield of wheat has been reported to decline by as much as 60% because of drought stress [6].

Adaptation of crops to water deficit is crucial, especially in a climate change scenario [7]. To tolerate abiotic stress, plants have many adaptation strategies. In this context, many studies focused on ways to secure food production in the future [8]. It was reported that plants cope with drought stress by developing drought avoidance and/or drought tolerance mechanisms, which include osmotic adjustment, regulation of stomatal conductance and photosynthesis, production of antioxidant and scavenger compounds, or regulation of water uptake and flow in their tissues [9,10].

It is clear that knowledge on crop responses to water deficit, and the mechanisms of drought tolerance is still limited [11]. Previous studies on drought tolerance in maize and other crops have shown tolerant cultivars with significant differences in antioxidant activity, presented as lower lipid peroxidation, improved accumulation of osmolytes and turgor adjustment, maintained photosynthetic activity and regulated enzymatic activities [11,12], whereas a few other studies focused on the effect of micro-organisms, like bacteria and fungi, on this constraint [13].

In this context, the symbiosis of arbuscular mycorrhiza (AM) fungi with plant roots has been shown to be helpful to tolerate and overcome water stress episodes in different plant species [14,15], including wheat [16]. It was previously reported that AM-plant association leads to better plant antioxidant activity, osmotic regulation, and root hydraulic properties [17]. Also, AM inoculated plants generally present a higher level of photosynthetic pigments, enhanced chlorophyll fluorescence traits and net photosynthetic rate [18], as well as different hormone regulation compared to control plants [19]. Symbiosis with mycorrhiza enhanced host plant tolerance to drought stress due to complex adaptations, including morphology, growth, and metabolism. The main consequences of this adaptation are a decrease of cell division and expansion, leaf size, stem elongation, and root proliferation, and disturbed stomatal oscillations, plant water, and nutrient relations [16,20].

It has been suggested that mycorrhiza colonization has a host-dependent plant colonization effect, which varies between plant species and genotypes [21]. Variation in plant response to AM inoculation exists for different wheat cultivars under drought stress [22]. The present study investigated the effect of AM inoculation on bread wheat cultivars under drought stress conditions for morphological, physiological, and oxidative metabolism changes. Therefore, the aim of this study was to determine the effect of AM inoculation on (a) the extent of regulation of various antioxidants and metabolites, and (b) the morphological traits in bread wheat cultivars in the presense and absence of drought stress. The hypothesis is that AM inoculation could improve drought tolerance in wheat. 

## 2. Results

### 2.1. Plant Growth

Throughout the growth period, drought stress differentially affected all traits. There was a significant increase in all the studied traits of every treatment, except for length of the last node for PAN3497 after 15 days of drought stress application. The decrease of length of the last node was around 70% for both treatments (T2 and T3). SST806 was more tolerant to drought stress than PAN3497 for all characteristics except for the length of the last node (Table 1). AM ameliorated all growth parameters of the two cultivars after 15, 30 and 45 days of stress application (T3). The highest amelioration was noted for leaf number after 45 days of stress, where the increase was 35% and 39% for PAN3497 and SST806, respectively. In addition, increase of tiller number due to mycorrhiza under drought stress was 38% and 45%, respectively (Table 1).

### 2.2. Chlorophyll Content

Drought stress (T2) reduced leaf chlorophyll content of both cultivars between 50% and 60% (Figure 1). Under the same conditions, this trait was also reduced in the AM treatment by 55.5% and 60%, respectively, for SST806 and PAN3497. Plants inoculated with AM exhibited the highest total chlorophyll content in both cultivars (Figure 1). Under drought stress conditions, plants inoculated with AM (T3) showed an increase of this trait compared to non-inoculated plants (T2). This increase was higher in PAN3497 (Figure 1). 

### 2.3. Proline Content

Drought stress (T2) increased proline content by 28% and 14%, respectively, for PAN3497 and SST806 (Figure 2). Under drought stress conditions, proline content decreased in the AM-treated cultivars (T3) (33 and 42% for PAN3497and SST806, respectively).

### 2.4. Oxidative Damage to Lipids

Drought stress conditions decreased the oxidative damage to lipids measured as malondialdehyde (MDA) equivalents significantly in both cultivars in comparison with control plants (25% and 70% respectively). The oxidative damage to lipids was significantly reduced by drought stress by 34% and 60% respectively, for PAN3497 and SST806 in AM-plants, (Figure 3a).

### 2.5. Total Soluble Sugars

The leaf total soluble sugar (TSS) concentration was significantly increased by the AM treatment in both cultivars (33% and 14% respectively for PAN3497 and SST806) under drought stress conditions (Figure 3b). The highest increase of 50% was seen in PAN3497. Plants under well-watered conditions did not alter their TSS content as consequence of the AM inoculation (Figure 3b).

### 2.6. Membrane Electrolyte Leakage and Water Deficit Saturation

Membrane electrolyte leakage (EL) increased by 30% and 36% under drought stress in PAN3497 and SST806, respectively. The AM treatment did not significantly affect this trait (Figure 4a). No significant variability was shown between cultivars due to drought stress (Figure 4a). The EL values were higher in non-AM drought-stressed plants (50% and 60% for PAN3497 and SST806) than in AM inoculated drought-stressed plants in both cultivars (40% and 45% respectively for PAN3497 and SST806) (Figure 4a). Drought stress did not increase water deficit saturation (WDS) in comparison with control plants (Figure 4b). AM inoculation increased WDS by 6% and 4%, respectively, in SST806 and PAN3497. AM did not affect WDS under drought stress.

### 2.7. Total Phenol Content

Total phenol content in the leaves of the two wheat cultivars under drought stress conditions (T2), was increased the most in SST 806 (44%). However, with AM inoculation (T3), this trait decreased compared to non-AM inoculation by 60% and 66% in PAN3497 and SST806, respectively, under drought stress conditions (Figure 4c).

### 2.8. Accumulation of Hydrogen Peroxide

The accumulation of hydrogen peroxide (H_2_O_2_) was significantly affected by AM inoculation (T1), increasing the values in AM plants under drought stress conditions (T3) (47% and 16% for PAN3497 and SST 806) (Figure 4d). Under drought stress conditions, H_2_O_2_ accumulation was higher in non-AM PAN3497 (70 mmol g^−1^) plants than in non-AM SST 806 plants (30 mmol g^−1^) (Figure 4d).

### 2.9. Peroxidase and Polyphenol Oxidase Activities

The peroxidase (PO) in leaves was on the order of 4 µmol g^−1^ FM in both cultivars under drought stress conditions (T2). With AM inoculation (T1), PO varied significantly by 30% and 37%, respectively, in PAN3497 and SST806 compared to the control treatment (T0) (Figure 5a). In general, drought stress (T2) did not significantly affect PO activity. However, the polyphenol oxidase (PPO) activity was increased under the same conditions (T2) (44% and 47%, respectively). It exceeded 16 µmol g^−1^ FM in the leaves of SST 806. In the AM-treatment, PPO was increased under stress conditions (T3) for both cultivars compared to the control (T1) (Figure 5b).

## 3. Discussion

The current study, along with previous studies [17] highlighted the divergent responses to AM inoculation of different bread wheat cultivars under drought stress conditions. Drought significantly decreased morphological and physiological traits in the two cultivars under both mycorrhiza inoculated and non-inoculated conditions [23]. Inoculation with AM enhanced the water absorption in many of the plants that were subjected to drought stress. AM fungus mycelia can penetrate a larger volume of soil than the host plant roots, therefore increasing water absorption and transport to the roots and other plant parts, resulting in reduced drought stress [24]. Inoculation with the AM fungus significantly influenced morphological traits, increasing mostly leaf and tiller number per plant. This confirmed the efficiency of the applied spore density and AM subspecies used in this study. These findings agree with those from a study [25] reporting a higher tiller number in mycorrhiza treated wheat. Another study [26] reported that many plants treated with AM had increases in different agronomic traits compared to those that were not treated.

In comparison with the plants that were not inoculated, the AM plants showed the highest increase of leaf number, which was 35% and 39% for PAN3497 and SST806, respectively. AM inoculation has been previously reported to enhance drought tolerance of host plants [27]. In the present study, the beneficial effect of the AM fungus under drought stress was observed in leaf and tiller number. Indeed, plant biomass production is an integrative index of plant performance under stress conditions and the efficiency of the AM symbiosis has often been measured in terms of host plant biomass improvement [9].

AM inoculation also improved photosynthetic capacity of leaves [28]. Total chlorophyll content was affected by drought stress. These results confirmed those of previous studies [29,30] where leaf photosynthetic capacity was negatively influenced by water deficit conditions, resulting in chloroplast dehydration. Regarding the effect of AM inoculation, the amount of *Chl a* and *b* in leaves was ameliorated in the AM-treated cultivars compared to those that were not treated. Higher concentrations of total *Chl* in inoculated cultivars indicated the positive effect of inoculation that stimulated increased photosynthesis under both stress and non-stress conditions, mostly in PAN3497.

Plants have evolved complex mechanisms allowing for adaptation to water deficits under drought stress. In this study, lower water deficit saturation of the inoculated plants under water stress conditions is probably due to increased water uptake mediated by mycorrhizal hyphae that can induce some cellular changes in plants. This agrees with previous findings [23]. In leaves, high EL was generally noticed under drought stress, accompanied by a significant increase of total phenol secretion and accumulation of H_2_O_2_ as evidence of plant physiological reaction to stress conditions.

The percentage of membrane EL, an indicator of cell membrane stability, has been identified as a good indicator of the tolerance to water stress [30,31]. Accordingly, non-AM inoculated drought-tolerant plants had lower EL values than the corresponding drought-sensitive ones. However, under drought stress, the two cultivars maintained variable values of EL, total phenols and H_2_O_2_. Interestingly, except for total phenol secretion, AM increased EL, WDS and H_2_O_2_ under drought compared to the controls in both cultivars. Similarly, it was reported [31,32] that AM symbiosis regulated different physiological mechanisms under drought stress in bread wheat. 

Plants also evolved to regulate compatible solutes like proline and TSS, termed as drought escape [33]. However, proline accumulation in drought stressed plants vary and depend on cultivar and treatment. AM inoculation increased this trait under drought stress (T3) compared to the AM inoculated treatment without stress (T1). It was also found that TSS concentration was significantly increased by the AM inoculation in both bread wheat cultivars under drought stress conditions. This agrees with previous findings that demonstrated that the AM fungi improve the plant osmotic adjustment by accumulation of different compounds (such as sugars, proline and free amino acids) [34,35]. This regulation by the AM symbiosis has been proposed as a mechanism allowing plants to grow under water stress [36]. In leaves of drought stressed plants, AM inoculated plants had increased TSS in both cultivars. The key effect of AM on sugar accumulation has been reported under drought conditions in various studies [19,36,37] as was also shown here in bread wheat. In addition, AM inoculation was shown as an alternative to escape drought stress conditions. In this sense, the higher membrane stability is often related to lower MDA levels [38] because of lipid peroxidation. These results agree with previous studies where MDA production was reduced by AM fungi [39]. Results in the current study showed that AM symbiosis decreased MDA accumulation under drought stress.

AM inoculated plants regulate some enzymatic activities as an adaptation strategy during drought stress episodes. These activities are stimulated in AM inoculated plants more than in non-inoculated plants [40]. Among these activities, this study focused on PO and PPO reaction under drought stress. In AM inoculated plants, the PO and PPO increased under stress conditions (T3) for both cultivars compared to the AM inoculated non-drought treatment (T1). These results confirmed those reported on AM-inoculated tomato under salt stress [41]. Activity of antioxidant defense enzymes was largely induced by AM-inoculation in tomato plants. Oxidative burst and generation of superoxide radicals occur during development of hypersensitive response in plant-stress interactions [42,43]. 

## 4. Materials and Methods

### 4.1. Soil and Biological Materials and Trial Layout

The experiment was carried out in a greenhouse at the University of Free State, Bloemfontein, South Africa. Two commercial, hard red spring wheat cultivars with excellent bread making quality and yield were used in the study (PAN3497 and SST806). These cultivars both have a medium growing period. They are the products of two different seed companies. Nothing is known about their root systems, but they are similar morphologically. The soil had a pH of 6.8 and comprised of 50% sand, 10% silt and 40% clay. Three seeds were planted per pot filled with 2 kg soil sterilized and fertilized with 7.5 mg kg^−1^ (P), 231.4 mg kg^−1^ (K), 564 mg kg^−1^ (Ca), 147.6 mg kg^−1^ (Mg). Mycorrhiza inoculum (commercial inoculum in powder form, registered and produced by Biocult (Pty) Ltd 005333/07, Somerset West, South Africa) was used as bio-inoculant as a seed application of 150 g per 150 kg of seed. Thirty minutes before sowing, seeds were mixed very well with the mycorrhiza inoculant. The active ingredient was mycorrizae subspecies, 400 spores per gram (as indicated by the manufacturer). The subspecies included *Glomus mosseae*, *Glomus intraradices*, *Glomus etunicatum*, and *Scutellospora dipurpurescens*. The experimental design was a randomized complete block design with two factors: (1) inoculation treatment and (2) drought stress application. The different combinations of these factors gave a total of four treatments for each cultivar which were non-inoculated control plants (T0), plants inoculated with the AM fungus (T1), non-inoculated plants under drought stress (T2) and inoculated plants under drought stress (T3). Three replicates were used for each treatment with 15 plants per replication, giving a total of 45 plants per treatment.

### 4.2. Growing Conditions

The experiment was conducted from May 2019 in the greenhouse. The growing period was 6 months under temperatures set to 19/25 °C night/day, with a relative humidity of 60–70%. Soil moisture was measured with a soil meter (Efekto Ltd., Caledon, South Africa). Water was supplied daily to maintain soil moisture at 100% field capacity during the first 4 weeks after sowing. At three-leaf stage, drought stress was applied and plants under stress were allowed to dry out until soil water content reached 25% field capacity, while the non-stressed treatments were maintained at 100% field capacity. The soil water content was measured daily with the soil meter before rewatering (late afternoon), reaching a minimum soil water content around 25% field capacity in the drought-stressed treatments [7].

### 4.3. Measurements

#### 4.3.1. Morphological Measurements

Morphological traits measured were plant length (cm), tiller number per plant, node number per tiller, length of the last node (cm) and leaf number per plant. Leaf length (cm) and leaf width (cm) were measured for all treatments before drought stress application and then 15, 30 and 45 days after drought stress application. Absorbance measurements for all treatments were done using a Jenway 7315 spectrophotometer (Cole-Parmer, Staffordshire, UK).

#### 4.3.2. Total Chlorophyll Content

Total chlorophyll content was measured using methods described previously [44,45]. It was extracted with acetone in a mortar, using 200 mg of fresh leaf tissue and 5 mL of acetone (80%, *v*/*v*). Total chlorophyll content was measured after centrifugation (10 min at 5000× *g*); the absorbance of the supernatant was measured at 663 and 645 nm. Chlorophyll concentration was calculated using the following formula Equation (1):Total Chlorophyll = 8.02 OD_663_ + 20.20 OD_645_.(1)

#### 4.3.3. Proline Content 

Proline content was measured using methods described previously [46,47]. It was extracted from leaf samples (100 mg FW) with 2 mL of 40% methanol:water. The tubes were incubated in a water bath at 85 °C for 30 min, then 1 ml of extract was mixed with 1 mL of a mixture of glacial acetic acid and orthophosphoric acid (6 M) (3:2; *v*/*v*) and 25 mg ninhydrin. After 1 h of incubation at 100 °C, the reaction was terminated in an ice bath to stabilize the purple color of the extract, then 5 mL of toluene was added to each tube and the absorbance of top purple aqueous layer was measured at 528 nm with a spectrophotometer. The proline concentration was determined according to the standard curve obtained using reference proline solutions.

#### 4.3.4. Total Soluble Sugar

Total soluble sugars were extracted from 100 mg of fresh leaf tissues in 100 mM potassium phosphate buffer (pH 7). Soluble sugars were analyzed from 0.025 mL of plant extract reacting with 3 mL of freshly prepared anthrone (200 mg anthrone, 100 mL 72% (*v*/*v*) H_2_SO_4_) and placed in a boiling water bath for 10 min. After cooling, the absorbance at 620 nm was determined. The calibration curve was drawn using glucose in the range of 0.2 to 0.4 mg mL^−1^ [48,49].

#### 4.3.5. Oxidative Damage to Lipids

Lipid peroxidation was estimated as the content of TBA-reactive substances (TBARS) and expressed as equivalents of MDA [50,51]. Lipid peroxides were extracted by grinding 100 mg of fresh leaf tissues with an ice-cold mortar and 6 mL of 100 mM potassium phosphate buffer (pH 7). Homogenates were filtered through one Miracloth layer and centrifuged at 15,000× *g* for 20 min. The chromogen was formed by mixing 200 mL of supernatants with 1 mL of a reaction mixture containing 15% (*w*/*v*) trichloroacetic acid (TCA), 0.375% (*w*/*v*) 2-thiobarbituric acid (TBA), 0.1% (*w*/*v*) butylated hydroxytoluene, 0.25 N HCl and by incubating the mixture at 100 °C for 30 min. After cooling at room temperature, tubes were centrifuged at 800× *g* for 5 min and the supernatant was used for spectro-photometric reading at 532 nm. The calibration curve was drawn using MDA in the range of 0.1–10 nmol.

#### 4.3.6. Water Deficit Saturation 

Water deficit saturation (WDS) was determined by weighing 100 mg (FW) young leaf tissue into a Petri dish and keeping it in distilled water (3 mL) for 24 h, thereafter weighing it again (turgid weight: TW). Finally, the leaves were dried for 72 h at room temperature and then weighed (dry weight: DW) again. WDS was calculated [47] as follows Equation (2):WDS (%) = (TW/FW)/(TW − DW) × 100(2)

#### 4.3.7. Electrolyte Leakage

Membrane permeability (electrolyte leakage, EL) was employed to evaluate the stability of the cell membrane. The EL and the electrical conductivity are related to the EC [47,52]. The electrolyte leakage is expressed as EL (%) = (EC1/EC2) × 100, where EC1 and EC2 denote the obtained values from incubated leaves at 25 °C for one day and that measured for autoclaved leaves at 120 °C for 20 min, respectively.

#### 4.3.8. Hydrogen Peroxide Content 

Hydrogen peroxide content was determined according to published methods [53,54]. A 100 mg of leaf sample aliquot was centrifuged at 15,000× *g* at 4 °C for 15 min after grinding it in 2 mL of TCA (20%). The supernatants of these extracts were collected to determine their H_2_O_2_ content. To 0.5 mL of the extract, 0.5 mL of potassium phosphate buffer (10 mM, pH 7) was added. One mL of potassium iodine (1 M) was added after incubating this for an hour in the dark. H_2_O_2_ content was expressed in µmol per g of fresh weight (FW). H_2_O_2_ content was determined using a standard curve established under the same conditions with known concentrations of the H_2_O_2_ range.

#### 4.3.9. Total Phenol Content

Total phenol content was measured on fresh leaves (100 mg) that were ground at 4 °C in 80% methanol using an ice-cold mortar. The homogenate was centrifuged for 3 min. Total phenol content was estimated on the supernatants based on the Folin-Ciocalteu method [55,56]. The absorption was determined at 760 nm. A calibration curve was constructed from freshly prepared solutions of (+)-catechin. The results were calculated as mg of catechin per g FW [56].

#### 4.3.10. Polyphenol Oxidase and Peroxidase Content

To prepare the enzyme extract of polyphenol oxidase (PPO) and peroxidase (PO), the reaction mixture consisted of 200 µL of H_2_O_2_ at 0.3%, 300 µL of guaiacol at 20 mM, 2 mL of phosphate buffer (0.1 M, pH 6), 1 mL of distilled water and 50 µL of enzymatic extract. PO activity was determined from the decomposition of H_2_O_2_ at 470 nm. For the PPO activity assay [54,57], the reaction mixture consisted of 500 µL catechol at 1.6% in phosphate buffer (0.1 M, pH 6), 250 µL of distilled water, 200 µL of phosphate buffer (0.1 M, pH 6) and 100 µL of enzymatic extract. Thereafter, the absorbance was recorded at 410 nm. For both enzymes, reading of the optical density was checked once every 60 s during the 3 min of incubation against a control, where the enzymatic extract was replaced by distilled water. Enzyme activities were expressed as the amount of protein decomposing 1 µmol of H_2_O_2_ per g FW.

### 4.4. Data Analysis

Analysis of variance (ANOVA) was performed using the SPSS statistical program v.13 (IBM Corporation, Armonk, New York, NY, USA) (http://oss.software.ibm.com/icu4j/ accessed on 1 February 2021), and subsequent comparison of means was done using Duncan’s multiple-range test at *p* = 0.05.

## 5. Conclusions

The current study showed that AM can contribute to protect plants against drought stress by alleviating the water deficit induced oxidative stress. Enhanced antioxidant enzyme activity and lower lipid per-oxidation with AM inoculation may contribute to better maintenance of the physiological reactions in plant under drought stress. AM contribution varied significantly between the two cultivars under drought stress conditions. Overall, SST806 was more tolerant than PAN3497 to drought stress. The reason for this is not clear, as these cultivars have similar morphological characteristics, but this should be investigated further. Further research is also needed to study the effect of bio-fertilization in response to drought stress under field conditions. It would also be interesting to compare the effects of inoculum containing different fungal species and spore concentrations.

## Figures and Tables

**Figure 1 plants-10-01756-f001:**
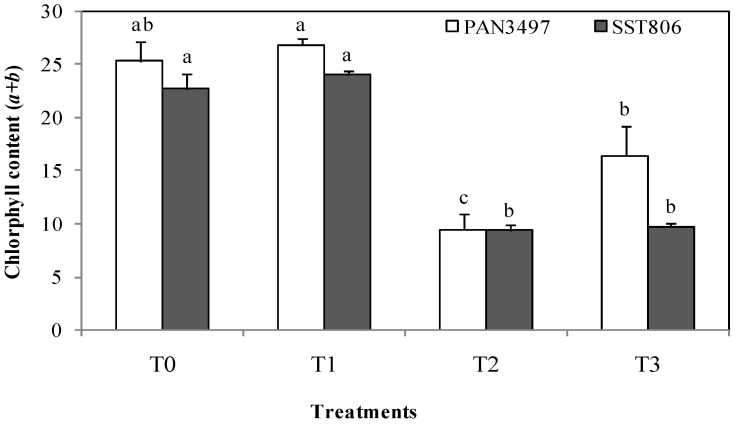
The effect of drought stress on leaf chlorophyll content of two bread wheat cultivars inoculated with arbuscular mycorrhiza and grown under greenhouse conditions. T0: non-inoculated control, T1: plants inoculated with the AM fungus, T2: non-inoculated plants under drought stress and T3: inoculated plants under drought stress. Bars with different letters are significantly different at *p* ≤ 0.05.

**Figure 2 plants-10-01756-f002:**
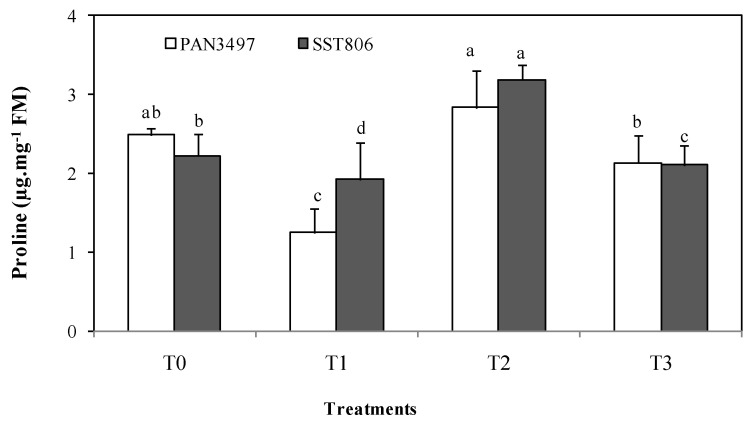
Effect of drought stress on leaf proline content for two bread wheat cultivars inoculated with arbuscular mycorrhiza and grown under greenhouse conditions. T0: non-inoculated control, T1: plants inoculated with the AM fungus, T2: non-inoculated plants under drought stress and T3: inoculated plants under drought stress. Bars with different letters are significantly different at *p* ≤ 0.05.

**Figure 3 plants-10-01756-f003:**
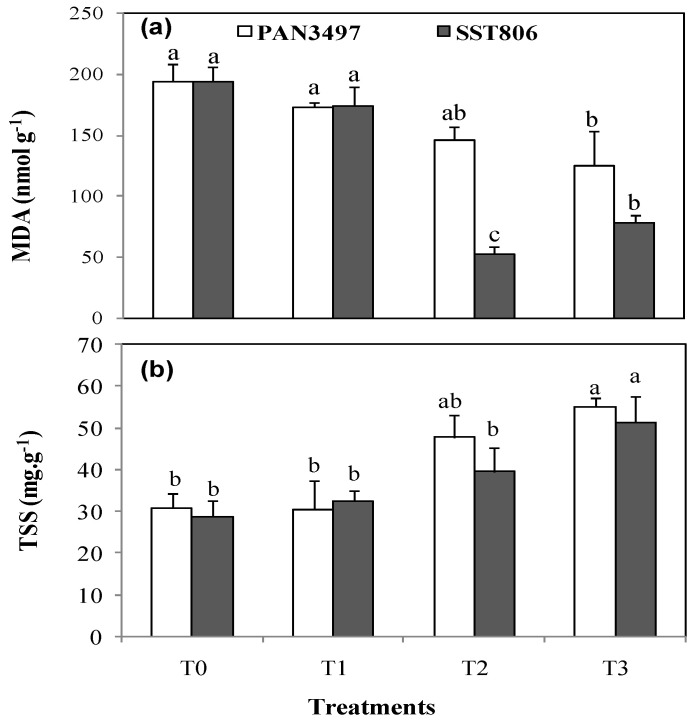
Effect of drought stress on malondialdehyde (MDA) (**a**) and total soluble sugar (TSS) (**b**) in the leaves of two bread wheat cultivars inoculated with arbuscular mycorrhiza and grown under green house conditions. T0: non-inoculated control plants, T1: plants inoculated with the AM fungus, T2: non-inoculated plants under drought stress and T3: inoculated plants under drought stress. Bars with different letters are significantly different at *p* ≤ 0.05.

**Figure 4 plants-10-01756-f004:**
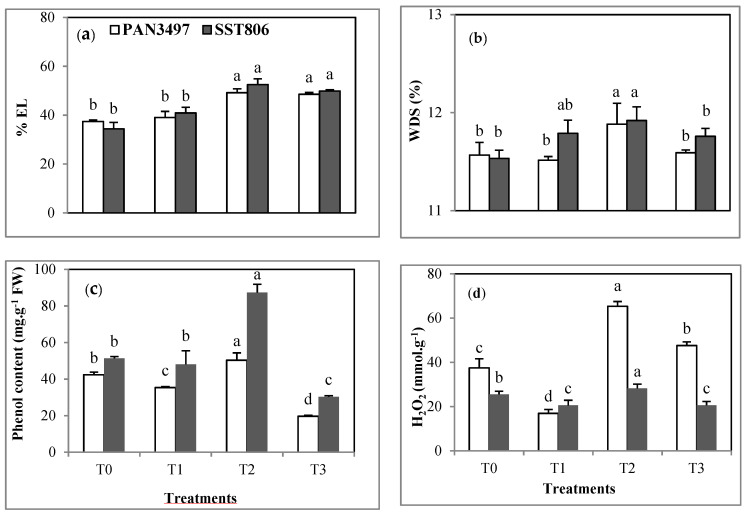
Effect of drought stress on: (**a**) membrane electrolyte leakage, (**b**) water deficit saturation, (**c**) total phenol content and (**d**) hydrogen peroxide accumulation in the leaves of two bread wheat cultivars inoculated with arbuscular mycorrhiza and grown under greenhouse conditions. T0: non-inoculated control plants, T1: plants inoculated with the AM fungus, T2: non-inoculated plants under drought stress and T3: inoculated plants under drought stress. EL: electrolyte leakage, WDS: water deficit saturation. Bars with different letters are significantly different at *p* ≤ 0.05.

**Figure 5 plants-10-01756-f005:**
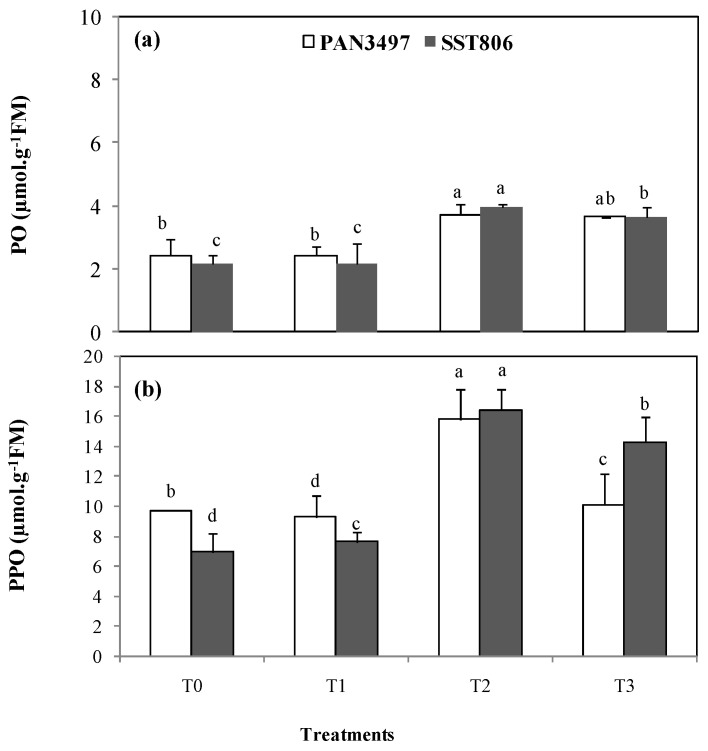
Effect of drought stress on peroxidase (PO) (**a**) and polyphenol oxidase (PPO) (**b**) activities in the leaves of two bread wheat cultivars inoculated with arbuscular mycorrhiza and grown under greenhouse conditions. T0: non-inoculated control plants, T1: plants inoculated with the AM fungus, T2: non-inoculated plants under drought stress and T3: inoculated plants under drought stress. Bars with different letters are significantly different at *p* ≤ 0.05.

**Table 1 plants-10-01756-t001:** Effect of drought stress on morphological characteristics in two bread wheat cultivars with different mycorrhizal treatments, grown under greenhouse conditions.

No stress application	**PL (cm)**			**TN Plant^-1^**		**NN Tiller^-1^**		**LLN (cm)**		**LN Plant^-1^**		**LL (cm)**		**LW (cm)**	
		**PAN3497**	**SST806**	**PAN3497**	**SST806**	**PAN3497**	**SST806**	**PAN3497**	**SST806**	**PAN3497**	**SST806**	**PAN3497**	**SST806**	**PAN3497**	**SST806**
	T0	23.06 ± 2.7 ^b^	21.92 ± 2.9 ^c^	1.46 ± 0.63 ^c^	1.53 ± 0.6 ^b^	1.66 ± 0.4 ^b^	1.2 ± 0.41 ^c^	1.44 ± 0.8 ^b^	1.71 ± 0.77 ^b^	3.93 ± 1.09 ^b^	4.6 ± 2.04 ^b^	20.14 ± 2.04 ^b^	18.21 ± 2.67 ^c^	0.58 ± 0.13	0.58 ± 0.15
	T1	28.60 ± 2.02 ^a^	25.4 ± 1.99 ^a^	2.26 ± 0.45 ^a^	1.8 ± 0.41 ^ab^	2 ± 0.53 ^a^	2.06 ± 0.45 ^a^	2.12 ± 0.60 ^a^	2.26 ± 0.66 ^a^	6.46 ± 1.40 ^a^	5.6 ± 1.05 ^a^	23.06 ± 1.7 ^a^	19.8 ± 1.8 ^a^	0.76 ± 0.11	0.71 ± 0.09
	T2	23.06 ± 1.6 ^b^	22.46 ± 0.87 ^c^	1.46 ± 2.05 ^c^	2 ± 0.23 ^a^	1.66 ± 0.48 ^b^	1.2 ± 0.42 ^c^	1.58 ± 0.76 ^b^	1.54 ± 0.72 ^c^	3.73 ± 1.27 ^b^	4.93 ± 1.90 ^b^	19.73 ± 1.89 ^c^	18.16 ± 1.77 ^c^	0.58 ± 0.10	0.58 ± 0.13
	T3	28 ± 1.2 ^a^	23.13 ± 2.4 ^b^	1.96 ± 1.98 ^b^	2 ± 0.50 ^a^	2 ± 0.38 ^a^	1.86 ± 0.40 ^b^	2.12 ± 0.66 ^a^	1.91 ± 0.66 ^ab^	6.06 ± 1.10 ^a^	4.86 ± 1.94 ^b^	23.06 ± 2.00 ^a^	18.73 ± 2.08 ^b^	0.76 ± 0.08	0.66 ± 011
15 days after stress application	**PL (cm)**			**TN Plant^-1^**		**NN**		**LLN (cm)**		**LN**		**LL (cm)**		**LW (cm)**	
		**PAN3497**	**SST806**	**PAN3497**	**SST806**	**PAN3497**	**SST806**	**PAN3497**	**SST806**	**PAN3497**	**SST806**	**PAN3497**	**SST806**	**PAN3497**	**SST806**
	T0	35.2 ± 3.9 ^a^	32.6 ± 2.7 ^a^	2.53 ± 0.63 ^b^	3 ± 0.84 ^b^	2.6 ± 0.50 ^b^	2.26 ± 0.45 ^c^	2.42 ± 1.2 ^b^	2.033 ± 0.97 ^a^	8.86 ± 1.7 ^b^	11.13 ± 3.5 ^ab^	26.13 ± 2.3 ^a^	25.46 ± 1.7 ^a^	0.86 ± 0.13 ^b^	0.86 ± 0.11 ^a^
	T1	34.53 ± 3.02 ^a^	31.37 ± 2.3 ^b^	3.46 ± 0.99 ^a^	3.4 ± 0.6 ^a^	2.46 ± 0.51 ^b^	2.73 ± 0.45 ^a^	2.67 ± 0.79 ^a^	1.63 ± 0.69 ^b^	11.33 ± 2.7 ^a^	11.86 ± 1.8 ^a^	26.06 ± 2.3 ^a^	25 ± 1.8 ^a^	0.97 ± 0.14 ^a^	0.78 ± 0.08 ^b^
	T2	27.86 ± 3.2 ^c^	25.93 ± 2.5 ^d^	1.66 ± 0.72 ^c^	2.06 ± 0.96 ^c^	2.8 ± 0.41 ^a^	2.4 ± 0.50 ^b^	0.73 ± 0.36 ^c^	1.47 ± 0.78 ^b^	5.93 ± 1.9 ^c^	7.26 ± 2.08 ^c^	22.06 ± 2.7 ^b^	21.13 ± 2.09 ^c^	0.62 ± 0.08 ^c^	0.70 ± 0.07 ^c^
	T3	28.8 ± 1.61 ^b^	28.53 ± 4.01 ^c^	1.66 ± 0.72 ^c^	2.8 ± 0.7 ^b^	2.8 ± 0.41 ^a^	2.26 ± 0.45 ^c^	0.72 ± 0.36 ^c^	1.95 ± 0.90 ^a^	5.93 ± 1.9 ^c^	9.66 ± 1.9 ^b^	22.06 ± 2.7 ^b^	22.2 ± 2.8 ^b^	0.62 ± 0.08 ^c^	0.72 ± 0.09 ^b^
30 days after stress application	**PL (cm)**			**TN Plant^-1^**		**NN**		**LLN (cm)**		**LN**		**LL (cm)**		**LW (cm)**	
		**PAN3497**	**SST806**	**PAN3497**	**SST806**	**PAN3497**	**SST806**	**PAN3497**	**SST806**	**PAN3497**	**SST806**	**PAN3497**	**SST806**	**PAN3497**	**SST806**
	T0	38.76 ± 2.5 ^a^	33.63 ± 1.63 ^b^	3.43 ± 0.86 ^b^	4.00 ± 0.92 ^c^	3.3 ± 0.25 ^a^	2.79 ± 0.51 ^b^	1.71 ± 0.65 ^bc^	2.38 ± 0.61 ^b^	13.93 ± 3.6 ^b^	17.23 ± 2.5 ^b^	27.06 ± 2.15 ^a^	24.89 ± 1.42 ^ab^	0.98 ± 0.11 ^a^	0.88 ± 0.10 ^b^
	T1	37.43 ± 2.76 ^b^	34.52 ± 3.17 ^a^	4.73 ± 0.99 ^a^	5.36 ± 1.55 ^a^	2.73 ± 0.25 ^c^	2.86 ± 0.22 ^b^	3.035 ± 1.04 ^a^	2.065 ± 0.64 ^c^	17.66 ± 4.1 ^a^	19.43 ± 4.4 ^a^	26.69 ± 2.3 ^ab^	25.5 ± 2.2 ^a^	1.02 ± 0.17 ^a^	0.91 ± 0.16 ^a^
	T2	29.76 ± 2.3 ^d^	28.96 ± 2.25 ^c^	2.16 ± 0.61 ^d^	2.53 ± 0.48 ^d^	3.40 ± 0.20 ^a^	3.2 ± 0.25 ^a^	1.86 ± 0.68 ^b^	1.8 ± 0.69 ^d^	9.8 ± 2.2 ^d^	10.63 ± 2.04 ^d^	22.03 ± 2.2 ^c^	21.9 ± 1.62 ^c^	0.71 ± 0.09 ^b^	0.78 ± 0.06 ^c^
	T3	31.73 ± 2.33 ^c^	31.76 ± 2.50 ^bc^	2.99 ± 1.11 ^c^	4.23 ± 0.63 ^b^	3.23 ± 0.49 ^b^	2.79 ± 0.22 ^b^	1.64 ± 0.80 ^c^	2.64 ± 0.83 ^a^	12.96 ± 4.55 ^c^	16.49 ± 1.2 ^c^	23.69 ± 2.5 ^b^	24.43 ± 3.15 ^b^	0.86 ± 0.08 ^b^	0.81 ± 0.09 ^b^
45 days after stress application	**PL (cm)**			**TN Plant^-1^**		**NN**		**LLN (cm)**		**LN**		**LL (cm)**		**LW (cm)**	
		**PAN3497**	**SST806**	**PAN3497**	**SST806**	**PAN3497**	**SST806**	**PAN3497**	**SST806**	**PAN3497**	**SST806**	**PAN3497**	**SST806**	**PAN3497**	**SST806**
	T0	42.33 ± 1.17 ^a^	34.67 ± 0.57 ^bc^	4.33 ± 1.1 ^b^	5.00 ± 1.00 ^c^	4.00 ± 0.00 ^a^	3.33 ± 0.57 ^b^	1.00 ± 0.10 ^d^	2.73 ± 0.25 ^b^	19.00 ± 5.5 ^bc^	23.33 ± 1.5 ^b^	28.00 ± 2 ^a^	24.33 ± 1.15 ^b^	1.10 ± 0.09 ^a^	0.90 ± 0.10 ^b^
	T1	40.33 ± 2.5 ^b^	37.67 ± 4.04 ^a^	6.00 ± 1.00 ^a^	7.33 ± 2.51 ^a^	3.00 ± 0.00 ^c^	3.00 ± 0.00 ^c^	3.40 ± 1.30 ^a^	2.50 ± 0.60 ^bc^	24.67 ± 5.5 ^a^	27.00 ± 7.00 ^a^	27.33 ± 2.3 ^b^	26.00 ± 2.6 ^ab^	1.07 ± 0.20 ^ab^	1.04 ± 0.25 ^a^
	T2	31.67 ± 1.5 ^d^	32.00 ± 2 ^c^	2.67 ± 0.5 ^c^	3.00 ± 0.00 ^d^	4.00 ± 0.00 ^a^	4.00 ± 0.00 ^a^	3.00 ± 1.00 ^b^	2.13 ± 0.60 ^c^	13.67 ± 2.5 ^c^	14.00 ± 2 ^c^	22.00 ± 1.7 ^d^	22.67 ± 1.15 ^c^	0.80 ± 0.10 ^b^	0.87 ± 0.05 ^b^
	T3	34.67 ± 3.05 ^c^	35.00 ± 1.00 ^b^	4.33 ± 1.5 ^b^	5.67 ± 0.57 ^b^	3.67 ± 0.57 ^b^	3.33 ± 0.57 ^b^	2.57 ± 1.25 ^c^	3.33 ± 0.76 ^a^	20.00 ± 7.2 ^b^	23.33 ± 0.5 ^b^	25.33 ± 2.3 ^c^	26.67 ± 3.5 ^a^	1.10 ± 0.09 ^a^	0.90 ± 0.10 ^b^

Numbers followed by different letters are significantly different at *p* ≤ 0.05. T0 = control, T1 = AM, T2 = drought stress, T3 = AM and drought stress. PL = plant length, TN = tiller number, NN = nodule number, LLN = length of last node, LN = Leaf number, LL = length of leaf, LW = Leaf width. Values in columns followed by different letters are significantly different at *p* ≤ 0.05.

## Data Availability

Data of this study is available from the authors.

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
