# Peer review of "Bread Wheat (Triticum aestivum) Responses to Arbuscular Mycorrhizae Inoculation under Drought Stress Conditions"

_plants, 2021, doi:10.3390/plants10091756_

Round 1

Reviewer 1 Report

The authors present a well-rounded study of the impact of mycorrhizal fungi on improving plant fitness using two bread wheat varieties as appropriate models. Overall, the paper is well written, however I have some specific comments for the authors to consider. 

Introduction: 

End of first paragraph - Reference No 6 is quoted to highlight the fact that drought is adversely affecting grain yield by as much as 60%. In a global context this is not the case, and the reference is looking at specific locations impacted. I strongly suggest you revise this statement. 

Results:

  1. The use of abbreviations from the beginning of the results section makes for very difficult reading, as this journal format requires materials and methods to be included in the latter part of the paper. The authors should give the full wording for the abbreviations in the results section, as it is the first mention of many of the abbreviations used.
  2. The authors while mentioning that the two spring wheat varieties, do not mention any characteristic features that may differentiate the cultivars. ie. are the root systems similar in size, differences in vigour, etc. As the authors demonstrate differences in plant response to AM fungi, greater emphasis should be made on identifying or as a minimum suggesting the biological mechanisms which may differ between the two cultivars.
  3. The more indicative data sets with respect to plant growth enhancement following inoculation ie. tiller number, leaf number and leaf width are best represented separately in graphs in my opinion for better visual appeal. 

Discussion: 

A more thorough discussion of the mechanisms surrounding the observed drought stress protective effects should be included (eg. Water absorption by AMF extraradical hyphae improving water availability to the plants resulting in reduced drought stress, upregulation of AMF antioxidant genes that protect the plant against free radical induced cellular damage,  etc.)

Author Response

Introduction: 

End of first paragraph - Reference No 6 is quoted to highlight the fact that drought is adversely affecting grain yield by as much as 60%. In a global context this is not the case, and the reference is looking at specific locations impacted. I strongly suggest you revise this statement. 

This was corrected.

Results:

  1. The use of abbreviations from the beginning of the results section makes for very difficult reading, as this journal format requires materials and methods to be included in the latter part of the paper. The authors should give the full wording for the abbreviations in the results section, as it is the first mention of many of the abbreviations us                                           This was corrected by describing the abbreviations in the results section.
  2. The authors while mentioning that the two spring wheat varieties, do not mention any characteristic features that may differentiate the cultivars. ie. are the root systems similar in size, differences in vigour, etc. As the authors demonstrate differences in plant response to AM fungi, greater emphasis should be made on identifying or as a minimum suggesting the biological mechanisms which may differ between the two cultivars.     These cultivars were selected as representatives of the two wheat breeding companies in South Africa, both with excellent yield and quality. No one has looked at their roots up to now, and for the rest they are very similar. So the reason for the difference between the two would be a separate study. We have included more information on the cultivars in the materials and methods section. 
  3. The more indicative data sets with respect to plant growth enhancement following inoculation ie. tiller number, leaf number and leaf width are best represented separately in graphs in my opinion for better visual appeal.     We considered this, but this would add a lot of pages to this paper and would make it too long, so we thought the best is to stick to the table format.  We hope you understand our point. 

Discussion: 

A more thorough discussion of the mechanisms surrounding the observed drought stress protective effects should be included (eg. Water absorption by AMF extraradical hyphae improving water availability to the plants resulting in reduced drought stress, upregulation of AMF antioxidant genes that protect the plant against free radical induced cellular damage,  etc.)

We have added to the discussion as suggested and we added a reference to support the added information. 

Reviewer 2 Report

Brief Summary

The manuscript plants-1326143 reports results on the capability of bread wheat seed treatment with arbuscular mycorrhizae to induce drought stress tolerance. A greenhouse experiment was set with two bread wheat cultivars (PAN3497 and SST806) under drought-stressed conditions in plants grown under greenhouse conditions. The effectiveness of the treatment was assessed by the estimation of growth parameters, lipid peroxidation, solute accumulation, water deficit saturation, photosynthetic activity, total phenol secretion and enzymatic activities (i.e., peroxidise and polyphenol oxidase. The results showed that seed treatment with arbuscular mycorrhizae improved morphological and physiological parameters in plants under drought stress. The manuscript topic is of great interest and the methods applied are appropriate; the data handling is suitable, and the findings reported considerable. However, the manuscript should be ameliorated for the English language and some other issues.

Broad comments

  • Title: I would revise the title by providing details of the vegetal species investigated and removing some elements that were not assessed (i.e. symbiosis). For example: “Bread wheat (or Triticum aestivum) seed treatment with arbuscular mycorrhizae for morpho-physiological parameters improvement under drought stress conditions.”
  • Abstract: Authors should improve this section by providing a clear aim of the work.
  • Introduction: The introduction places the study in a broad context, introducing its importance. However, in this section, the aim of the study is once again not clearly stated. Moreover, the specific hypotheses being tested should be provided.
  • Results: The results description is clear. To emphasize the results got I would also provide the chlorophylls a and b ratio, usually affected by drought stress and microbial inoculation.
  • Materials and Methods: The authors described very well the experimental design. However, the other methods that were used for measurements were described too concisely. Please provide brief methods description in separated sub-headings for each measurement.
  • Discussion: The authors discussed too briefly the results from the perspective of previous studies and the purpose of the study. The authors completely missed a discussion related to the microbial composition of the product tested in their work. Have the studies cited a similar fungal composition to the product tested? Is the composition tested more effective because of mixed fungal species or their spore density in the product?
  • Conclusions: The section is appropriate as the study contains many elements. In this section, the authors should also highlight the strength and limitations of the work and present some details of future research directions mentioned.
  • References: References should be revised (journal abbreviations correctness and punctuation).

Specific comments

Figure 3 – The quality of this figure should be ameliorated.

Figure 5 – To improve the readability of the title of the Y-axis I suggest using PP and PPO abbreviations. The scale of the Y-axis of Figure 5a should be rescaled.

L315 – Please, provide a brief description of product composition (fungal species and active and inert ingredients).

L318 - How this potential was determined?

For the English language, some sentences are too long and there are some typos (e.g. L. 317, inocculant). Please read the complete manuscript again.

Author Response

  • Title: I would revise the title by providing details of the vegetal species investigated and removing some elements that were not assessed (i.e. symbiosis). For example: “Bread wheat (or Triticum aestivum) seed treatment with arbuscular mycorrhizae for morpho-physiological parameters improvement under drought stress conditions.”
  • We have changed the title to: Bread wheat (Triticum aestivum) responses to arbuscular mycorrhizae inoculation under drought stress conditions
  • We hope this is acceptable. 
  • Abstract: Authors should improve this section by providing a clear aim of the work.
  • We have added an aim to the abstract
  • Introduction: The introduction places the study in a broad context, introducing its importance. However, in this section, the aim of the study is once again not clearly stated. Moreover, the specific hypotheses being tested should be provided.
  • We have added the aims of the study and a hypothesis to the introduction.
  • Results: The results description is clear. To emphasize the results got I would also provide the chlorophylls a and b ratio, usually affected by drought stress and microbial inoculation.
  • As we measured so many traits and covered so many physiological aspects, we did not separate the a and b chlorophylls but simply used them combined. We hope this will be sufficient.  
  • Materials and Methods: The authors described very well the experimental design. However, the other methods that were used for measurements were described too concisely. Please provide brief methods description in separated sub-headings for each measurement.
  • We have added a sub-heading for each of the methods used in laboratory analysis, and then described each method.. 
  • Discussion: The authors discussed too briefly the results from the perspective of previous studies and the purpose of the study. The authors completely missed a discussion related to the microbial composition of the product tested in their work. Have the studies cited a similar fungal composition to the product tested? Is the composition tested more effective because of mixed fungal species or their spore density in the product?
  • We have managed to find the composition of the inoculum used and we added this information to the materials and methods section. The manufacturer just listed the contents. We used the dose prescribed by the manufacturer. So we looked at the influence of this available product at the rate prescribed to producers, and we did not scrutinize the precise mixture of fungal species or the spore density.  This would be a very interesting follow-up study. 
  • Conclusions: The section is appropriate as the study contains many elements. In this section, the authors should also highlight the strength and limitations of the work and present some details of future research directions mentioned.
  • We have added information to this effect to the end of the conclusion section. Here we specifically referred to the issue of the composition of the inoculum and the fungal spore concentration which should be looked at in a follow-up. 
  • References: References should be revised (journal abbreviations correctness and punctuation).
  • We checked these again and corrected.

Specific comments

Figure 3 – The quality of this figure should be ameliorated.

Figure 5 – To improve the readability of the title of the Y-axis I suggest using PP and PPO abbreviations. The scale of the Y-axis of Figure 5a should be rescaled.

Both these figures were improved in the revision. 

L315 – Please, provide a brief description of product composition (fungal species and active and inert ingredients).

We have added this information in the materials and methods section. The manufacturer could supply us with the subspecies, which are listed as the active ingredient. 

L318 - How this potential was determined?

The manufacturer provided this information. 

For the English language, some sentences are too long and there are some typos (e.g. L. 317, inocculant). Please read the complete manuscript again.

We tried to correct all language issues.

Reviewer 3 Report

Well written document, everything is clearly explained.

Only two-three corrections and one question: What about the impact on yield? Do you have data on number of grains/ear and thousand kernel weight for each treatment?

Author Response

We have made all the corrections as indicated on the pdf file. There was one question regarding the yield components. We did not include it in this paper, but will include that in a different paper. 

Round 2

Reviewer 2 Report

The authors correctly addressed all my previous comments. I have no further suggestions.